# Online Health Information-Seeking Behaviours and eHealth Literacy among First-Generation Chinese Immigrants

**DOI:** 10.3390/ijerph20043474

**Published:** 2023-02-16

**Authors:** Ling Zhang, Sherrie Chung, Wendan Shi, Dion Candelaria, Robyn Gallagher

**Affiliations:** 1Susan Wakil School of Nursing and Midwifery, Faculty of Medicine and Health, University of Sydney, Camperdown, NSW 2006, Australia; 2Charles Perkins Centre, University of Sydney, Camperdown, NSW 2006, Australia

**Keywords:** health literacy, eHealth literacy, digital health literacy, access to information, cultural diversity, ethnicity, emigrants and immigrants, vulnerable populations

## Abstract

Due to linguistic and cultural barriers, immigrants often have limited access to health information. Online health information is popular and accessible, but quality is questionable and its benefits dependent on an individual’s eHealth literacy. This study examined online health information-seeking behaviours, eHealth literacy and its predictors among first-generation Chinese immigrants. A sample of 356 Chinese immigrants living in Australia completed an anonymous paper-based survey, including sociodemographic, clinical data, English proficiency, health literacy, online health information-seeking behaviours, and eHealth literacy. Linear regression models analyzed predictive factors of eHealth literacy. Participants were aged mean 59.3 years, female (68.3%), 53.1% completed university, and their English proficiency was rated fair/poor by 75.1%. Participants perceived online health information as useful (61.6%) and important (56.2%) to their health. Health information accessed was often related to lifestyle (61.2%), health resources (44.9%), diseases (36.0%), and medications (30.9%). Inadequate health literacy and eHealth literacy occurred in 48.3% and 44.9%, respectively. Age, number of technological devices used, education, and health status were independently associated with eHealth literacy. While most Chinese immigrants used online health information, many had inadequate eHealth literacy. Healthcare authorities and providers should support older immigrants, those with lower education and poorer health, and those less engaged with technology in online health information use by providing culturally and linguistically appropriate information, directing immigrants to credible websites, and involving them in health material development processes.

## 1. Introduction

Global migration has accelerated rapidly over the past two decades, with an estimated 281 million international migrants living outside of their country of origin in 2020 [1]. Immigrant integration is essential to various aspects of an immigrant’s life, including health, and a key driver for this process is language proficiency [1]. Despite the importance of language acquisition, proficiency levels in the host country’s language remain low for large segments of the immigrant population [2,3]. Major migrant-receiving countries, such as the United States, Canada, and Australia, report that between 37% to 68% of immigrants have poor language proficiency, and many never reach adequate levels in the host country’s language [4,5,6]. Those immigrants who migrated at an older age are at particular risk [7,8].

The influence of language proficiency extends beyond an individual’s social and economic participation. For instance, inadequate language proficiency directly leads to poor access to health information, health services, and communication with healthcare providers [9,10,11,12]. Additional information access barriers are related to cultural disparities between immigrants and healthcare providers in the host countries, and socio-economic disadvantages among migrant populations [13,14]. Therefore, immigrants commonly seek alternative and accessible health information in their own languages [15,16]. Emerging evidence shows that online health information has become one of the most popular alternative sources for Chinese immigrants [15,17].

However, the benefits of online health information depend upon individuals’ eHealth literacy, that is, their capabilities for finding, understanding, appraising, and applying online health information to their daily health-related decisions [18,19,20]. The variable quality of online health information caused by inadequate regulation makes it even more difficult for people without adequate critical eHealth literacy to use the information obtained. Non-credible health information including misleading or wrong health information or those purposely used for promoting products can have negative influences or even be harmful to those with limited skills to differentiate good and bad quality information [17,19,20].

Chinese immigrants, who were born in mainland China, Hong Kong, Macau, and Taiwan and are living outside of their country of origin, are one of the largest immigrant populations worldwide [21]. The available literature shows that this population often exhibits language issues in host countries with three-quarters reporting having inadequate English proficiency [17,22,23,24,25] while also holding different health beliefs [11]. While the Chinese immigrant population is known to use online health information [17], and has high smartphone ownership and unique health information needs [26,27], the knowledge of their eHealth literacy and associated factors is lacking. An understanding of eHealth literacy skills, particularly in relation to internet use, the most common source of health information currently, will ensure that the design, development, and provision of online health information are appropriate for Chinese immigrants’ needs. Health communication strategies will be modified to address any gaps in understanding of how and what to use in terms of health information on the internet and guide health professionals to direct this population appropriately. Hence, the aim of this study was to address the knowledge gap by examining online health information-seeking behaviours, eHealth literacy, and associated predictors among first-generation Chinese immigrants living in Australia.

## 2. Materials and Methods

This was a survey study which was approved by the University of Sydney Human Research Ethics Committee (protocol number 2017/335) and followed the ethical principles of the World Medical Association outlined in the Declaration of Helsinki [28]. The study recruitment process is illustrated in Figure 1.

### 2.1. Setting and Sample Criteria

Participants were recruited from July to October 2017 from Chinese community organisations based in suburbs with a high proportion of Chinese immigrants in metropolitan Sydney, Australia. Participants were eligible if they (1) were born in Mainland China, Hong Kong, Taiwan, or Macao and were living in Australia at the time of the study; (2) spoke and understood sufficient Mandarin in verbal and written form for consent and survey processes; and (3) were aged ≥18 years. Participants were excluded if they reported a neurocognitive disorder such as dementia, Alzheimer’s disease, or major stroke.

The Raosoft sample size calculator was used to calculate the sample size. A total of 377 participants was required to obtain a representative sample, with a 5% margin of error and a 95% confidence level in a population of >20,000. An additional 10% recruitment was undertaken to allow for missing data.

### 2.2. Procedure

The leaders of not-for-profit Chinese community organisations across metropolitan Sydney were contacted and provided with study information by a bilingual researcher (LZ). These community leaders then distributed the study information to their members during community events. Community members who provided oral or written consent were provided with paper-based survey questionnaires to complete and return to the researcher at that time. The researcher was on site to assist with any explanations needed. Community members were encouraged to pass on the study information to their family members and friends, who were also then invited to participate.

### 2.3. Data Collection

Self-reported data collection occurred using a paper-based survey, which included sociodemographic data (age, gender, years of living in Australia, English proficiency, living arrangements, education achieved, and employment status) and clinical data (current diagnoses, comorbidities, and health status). These sample characteristics are the factors known to be related to health literacy and eHealth literacy. English proficiency and health status were reported on a 5-point Likert scale ranging from ‘Excellent’, ‘Very good’, ‘Good’, ‘Fair’, to ‘Poor’, which is used by Australian Bureau of Statistics for population level health data [29], and therefore is appropriate for the sample of this study. Online health information- seeking behaviours (health resource, disease, lifestyle, medication), and the type of technological devices used (desktops, laptops, smartphones, mobile phones, activity trackers) were also collected using a checklist, which was developed by the research team for previous studies [30].

### 2.4. Health Literacy

Participants’ health literacy levels were measured using a single health literacy screening question: ‘How confident are you filling out medical forms by yourself?’ [31]. Participants responded on a 5-point Likert scale (‘Aways’ to ‘Never’) with scores ≥3 considered inadequate/marginal health literacy. The health literacy screening question is a validated tool to differentiate people with adequate and inadequate/marginal health literacy, and it has been used in multiple studies [31,32].

### 2.5. eHealth Literacy

Participants’ eHealth literacy was measured using the eHealth Literacy Scale (eHEALS), which is a self-reported questionnaire that assesses an individual’s combined knowledge, comfort, and perceived skills at finding, evaluating, and applying electronic health information to health problems [33,34]. The eHEALS consists of ten items, with the first two items surveying individuals’ perceptions of the usefulness and importance of online health resources concerning their health. These two questions are measured on a 5-point Likert scale for usefulness (‘Not useful at all’ to ‘Very useful’) and importance (‘Not important at all’ to ‘Very important’). A further eight items measure an individual’s eHealth literacy level (Box 1) and the respondents generate a rate based on a 5-point Likert scale from ‘Strongly disagree’ to ‘Strongly agree’ [34]. Possible scores range from 8 to 40, with higher scores indicating greater perceived skills in using online health resources for health. A score of <26 is considered inadequate eHealth literacy. The eHEALS has been validated and used in many studies among populations with different conditions or speaking different languages [33]. For the current study, the survey was translated into simplified Chinese by bilingual researcher LZ, and back-translated into English by another bilingual researcher.

Box 1eHEALS items.
1.I know what health resources are available on the Internet2.I know where to find helpful health resources on the Internet3.I know how to find helpful health resources on the Internet4.I know how to use Internet to answer my questions about health5.I know how to use the health information I find on the Internet to help me6.I have the skills I need to evaluate the health resources I find on the Internet7.I can tell high quality health resources from low quality health resources on the Internet8.I feel confident in using information from the Internet to make health decisions


### 2.6. Data Analysis

Data were analysed using IBM SPSS version 26 [35]. Descriptive data are presented as mean, standard deviation (SD), frequency, and percentage as appropriate. Independent samples t-tests were used to determine differences in eHealth literacy according to age (<65/≥ 65 years), gender (male/female), years in Australia (<5/≥5 years), English proficiency (poor-fair/good-excellent), education (university level/<university), employment (employed/unemployed), living arrangement (lives alone/not alone), health status (poor-fair/good-excellent), technology use (≤1/≥2 devices), and health literacy (adequate/inadequate and marginal). A linear regression model was used to determine the independent associates of eHealth literacy using age, gender, education, English proficiency, technology use, health literacy, and self-reported health status. Statistical significance was set at *p* < 0.05 for all tests. Participants with missing data on important variables such as eHEALS items were excluded from the analysis.

## 3. Results

### 3.1. Participant Characteristics

In total, 415 surveys were received, of which 59 were incomplete and could not be used. Participants (*n* = 356) had a mean age of 59.3 (SD 16.0) years (Table 1). The majority of participants were female (*n* = 243, 68.3%) and had lived in Australia for 12.8 (SD 9.2) years at the time of the study. More than half of the participants had completed university (*n* = 189, 53.1%) and the majority were not in the workforce (*n* = 233, 66.1%). Only 24.9% (*n* = 90) rated their English proficiency as good to excellent. About half of the participants had inadequate/marginal health literacy (*n* = 172, 48.3%). Participants reported their health status as fair or poor (*n* = 206, 57.2%) and they had one or more chronic conditions (*n* = 256, 72.1%), the most common of which were arthritis (*n* = 106, 29.9%), back pain (*n* = 64, 18.0%), cardiac conditions (*n* = 57, 16.1%), and diabetes (*n* = 44, 12.4%).

Smartphones were the most common devices participants owned (*n* = 316, 88.8%) followed by tablets (*n* = 168, 47.2%), laptops (*n* = 132, 37.1%), and desktop computers (*n* = 118, 33.1%) (Figure 2). Most of the participants used the internet for health information (*n* = 281, 78.9%) and perceived online health information to be useful (*n* = 219, 61.6%) and important (*n* = 200, 56.2%) in managing their health. The online health information that was sought related to lifestyle (*n* = 218, 61.2%), health resources (*n* = 160, 44.9%), disease (*n* = 128, 36.0%), and medications (*n* = 110, 30.9%).

### 3.2. eHealth Literacy

eHEALS scores indicated 44.9% (*n* = 160) of participants had an inadequate eHealth literacy (score < 26). The mean eHealth literacy score was 24.8 (SD 8.0) and scores ranged from 8 to 40 (IQR: 21–31).

Respondents scored highest for items related to knowing the type of health resources available on the internet and finding this information useful (Table 2). For example, 51.3% of participants agreed on ‘I know how to use the health information I find on the Internet to help me’ and 46.9% of participants agreed on ‘I know what health resources are available on the Internet’, whereas the lowest mean scores were related to having the skills needed to evaluate the health resources they found and feeling confident to use this information for their health. For example, only 37.6% of the participants agreed on ‘I feel confident in using information from the Internet to make health decisions’ and 39.2% of the participants agreed on ‘I have the skills I need to evaluate the health resources I find on the Internet’.

The subgroups with the lowest eHealth literacy scores were older (≥65 years) (*p* < 0.001), male (*p* = −0.02), unemployed (*p* < 0.001), reported poor health status (*p* < 0.001), did not have a university education (*p* < 0.001), had poor/fair English proficiency (*p* < 0.001), had inadequate/marginal health literacy (*p* < 0.001), and used one or no technological devices (*p* < 0.001) (Figure 3).

The independent associates of eHealth literacy were determined using linear regression (Table 3). The model was significant (R = 0.578, R^2^ = 0.334, adjusted R^2^ = 0.327, *p* < 0.001). After adjusting for gender, English proficiency level and health literacy, eHealth literacy was worsened with increasing age (B = −0.093, 95% CI = −0.154, −0.031) and poorer self-reported health status (B = −1.544, 95% CI = −2.592, −0.496), and better with increasing number of technological devices used (B = 1.103, 95% CI = 0.404, 1.801) and having a higher education level (B = 3.088, 95% CI = 1.753, 4.423) (Appendix A).

## 4. Discussion

In this sample of Chinese immigrants living in metropolitan Sydney, Australia, who had multiple health conditions, a substantial proportion had inadequate eHealth literacy despite seeking online health information frequently. Most of the participants owned smartphones and actively used online health information and perceive it as useful and important for their health. Chinese immigrants most at risk of inadequate eHealth literacy were older, used technology less, were less educated, and had poorer health status. Despite high education levels and long residency in Australia in this sample, a large proportion reported having poor English skills. The phenomenon could be due to a large portion of Chinese immigrants recruited in the study who migrated at a relatively older age. Many of them had little or no exposure to the English language during their schooling and university education and possibly lived a segregated life in Australia. The study helps to understand eHealth literacy, access to technology, and the health information-seeking behaviours of Chinese immigrants, which is an important step in creating mHealth messages, programs, and interventions to improve the health of and prevent disease for this population. 

Many of the Chinese immigrants in this study had inadequate eHealth literacy. Their overall eHEALS mean scores (24.8) and item scores were lower than other ethnic minorities living in Western countries who had reported mean scores ranging from 28.1 to 30.4 [36,37,38]. The disparities may be due to a comparatively older age of the participants in this study than the other studies. Age is a proven predictor of poorer eHealth literacy skills in many studies across various populations regardless of ethnicity or migration status [7,8,38]. However, the patterns of the item responses across ethnic minority groups are very similar, as consistently demonstrated across ethnic minorities from China, south Asia, Africa, and the Middle East [18,34]. The areas with the most deficit include distinguishing the quality and evaluating health information or using the information when considering their health choices [13,20], which are beyond functional or interactive health literacy. These critical health literacy skills often require advanced personal skills including health knowledge, effective interaction between service providers and users, informed decision making, and empowerment [39]. Studies show that people with good critical health literacy skills are less vulnerable to online information exploitation [23,25], and also more likely to make appropriate health-related decision independently, have better self-efficacy, and engage in optimised self-care in chronic conditions [40,41]. Given the chronic conditions this sample of Chinese immigrants had, population approaches focusing on improving critical eHealth literacy skills will be beneficial.

eHealth literacy scores varied across subgroups within the study sample of Chinese immigrants, with the most at-risk groups being older, those who use fewer technological devices, and have a lower education level and poorer health status. These findings are consistent with the findings from studies in both ethnic minorities and non-ethnic minorities that show age, education especially university level education, and health status are the predictors of eHealth literacy [36,38,42]. This is because age, education, and health status are strongly associated with technology and internet exposure and adaptation, as shown in a large body of the literature [36,37,38,43], which is very likely to consolidate the skills needed to access online health information. However, the predictors of eHealth literacy vary across ethnic groups. For example, Makowsky et al. [36] reported significantly lower eHEALS scores in older people and those without university or college education and living with a chronic condition, although they did note that male gender is also at risk, unlike in this study. James et al. and Bergman et al. [37,38] further confirm that those who do not own a technological device or use the internet less frequently show significantly lower overall eHEALS scores. However, English language proficiency was not independently associated with eHealth literacy scores in this study which is inconsistent with Makowasky et al.’s and Bergman et al.’s studies [36,37]. In both studies, language-related factors were the predictors of eHEALS scores [36,37]. These differences could have arisen from the difference in measurements in the two studies, but further investigations are needed to determine the association between language proficiency and eHealth literacy among ethnic minorities or immigrants.

The positive perception of internet health information, active online information-seeking behaviours, and high ownership of technological devices among the study sample pose opportunities for internet or digital technology-based interventions in this population. Although health-related internet use is multifactorial, perceived usefulness and importance is one of the key prerequisites of health-related internet use [44]. Furthermore, high ownership of technological devices, especially smartphones, and existing online health information-seeking behaviours indicate a certain level of integration of technology into the daily lives in this population, which has proven to be one of the important predictors of ongoing adherence to online interventions [45].

Despite all these positive aspects of health-related internet use, online opportunities should be considered against the background of overall low eHealth literacy scores of this population, especially for eHealth literacy and the subgroups with the most difficulty, as well as the quality of online health information. At present, anyone can publish health information online and there is no regulation of quality control. Previous research on the content of health-related websites has highlighted inaccuracies that raise concerns about the quality of the online health information encountered by people [46]. Moreover, even the information translated by health authorities to provide locally relevant information for immigrants does not always achieve optimal quality [17]. On the other hand, immigrants often take years or even longer to integrate into the receiving societies, so there is a critical period of time before immigrants learn the language and the healthcare system of host countries [13,23]. During this critical period, guidance and support to direct and facilitate using credible online health information are critical in the prevention of online information exploitation. For example, health authorities and healthcare providers should provide accurately translated and culturally adapted online health information, recommend credible websites for immigrant populations, and involve community members in health material development to bridge the language and digital divide.

Moreover, it is important to ascertain when answering the eHEALS if the Chinese immigrants related their answer to information in English or to information from their country of origin. It is therefore important to link future research on eHealth literacy with evidence from information literacy to broaden the understanding of how people seek out and use information in everyday life [47]. Further, the validity of the eHEALS, and the relationships between eHealth literacy, health, and health behaviours are important and worth further investigation.

### 4.1. Strength

This study has a representative sample from right across metropolitan Sydney, Australia with varying ages, education and English proficiency levels, and health status. Additionally, data were collected on a paper-based format rather than an online survey, which maximises the inclusion of those less engaged with the Internet or technology. The findings of the study provides useful insight in designing or educating online health information use among Chinese immigrants, especially in the post-COVID world where Internet health information is becoming so accessible and inevitable.

### 4.2. Limitations

The study has several limitations including that the study sample was recruited from community organisations in one city, which could limit the generalizability of the findings to the broader Chinese immigrant population. Furthermore, self-reported data including language proficiency and health literacy are subject to social desirability bias. The primary variable of interest was eHealth literacy, and few measurement tools were available to choose from, particularly in simplified Chinese [48]. eHEALS is the most commonly used eHealth literacy tool and applies only to those with the skills to use Internet health resources rather than broad digital information sources. The research team has taken all proper measures to maintain the accuracy and clarity of the translation of the tool; however, the results should be interpreted with caution.

### 4.3. Implications

The results of this study indicate strong potential for internet-based health interventions for Chinese immigrants. However, more support is needed for the many people with inadequate eHealth literacy, especially in the areas related to appraisal and applying online health information to health and in elderly, less educated, less technologically capable, and health illiterate groups.

## 5. Conclusions

The study findings demonstrate that many Chinese immigrants living in the community consider online health information as both important and useful and use online health information frequently. This is important information indicating that health communication strategies about many conditions should be provided on the internet by health authorities. However, the study findings also demonstrate that a substantial proportion do not have the eHealth literacy skills required to evaluate the quality of the information or the confidence to apply the information obtained for their health situation. This is especially the case for immigrants who are older, less educated, unfamiliar with a variety of technologies, or have a poorer health status, creating difficulties with using online health information. Given that many Chinese immigrants live with chronic conditions and lack English language proficiency, online health information is likely the most accessible resource for this population. Therefore, healthcare authorities and providers should focus on directing immigrants to credible and useful websites appropriate for their condition. Additionally, we recommend that health providers develop culturally and linguistically appropriate health information for the common chronic conditions and engage immigrant consumers in the designing of the online materials to ensure acceptability and utility. Future research should determine the validity of eHEALS for this immigrant population and others and investigate the relationship between eHealth literacy and actual health behaviours undertaken in relation to the health condition of the populations.

## Figures and Tables

**Figure 1 ijerph-20-03474-f001:**
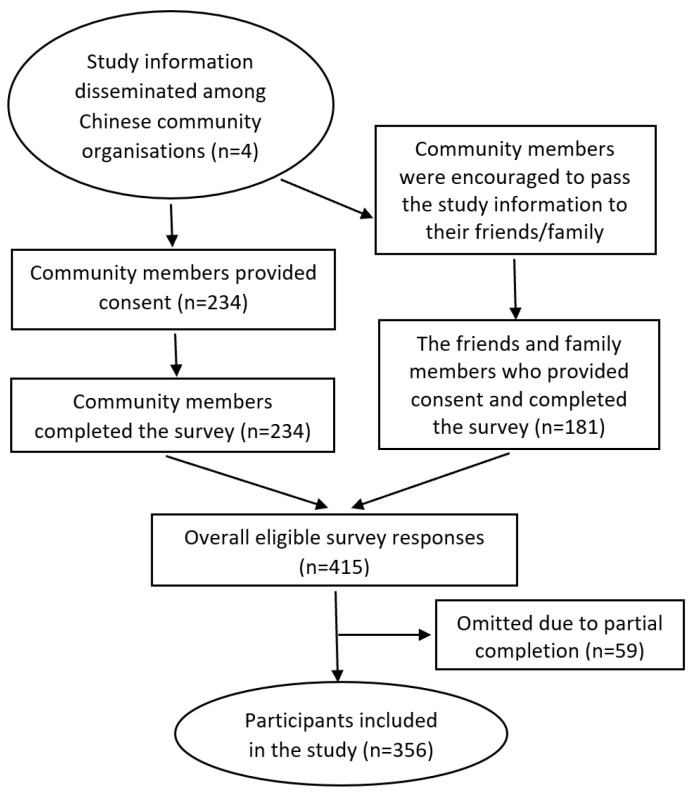
Study recruitment diagram.

**Figure 2 ijerph-20-03474-f002:**
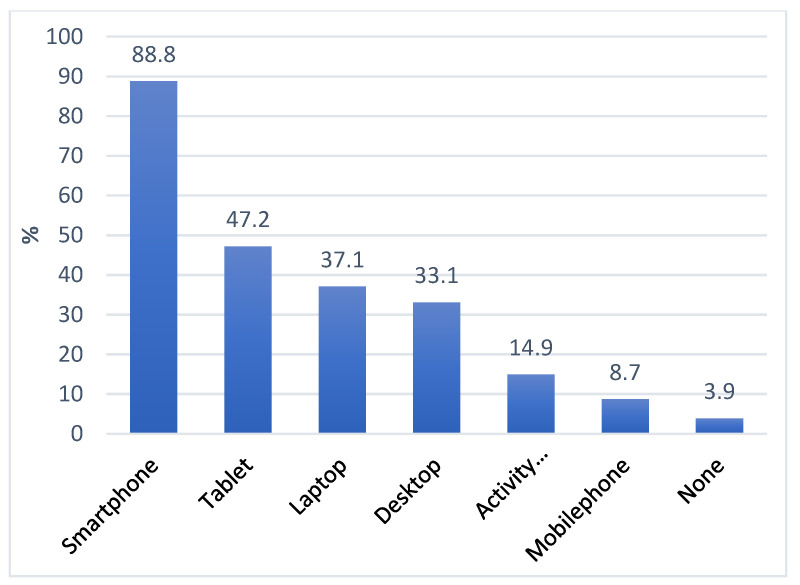
Ownership of technological devices among Chinese immigrants.

**Figure 3 ijerph-20-03474-f003:**
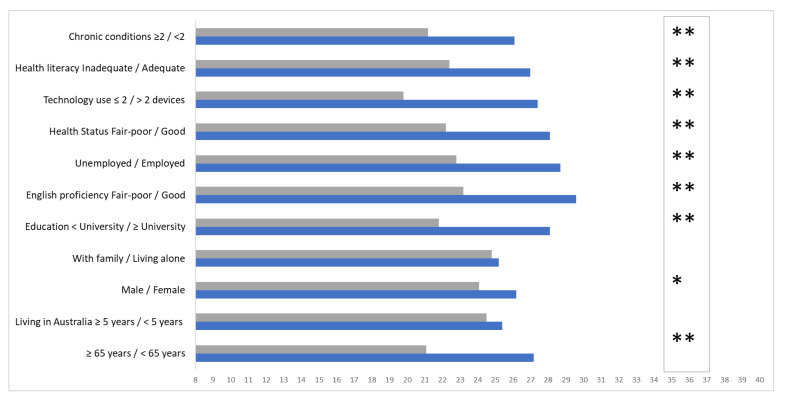
eHealth literacy scale scores compared by sociodemographic sub-groups. * Indicates *p* < 0.05, ** indicates <0.001.

**Table 1 ijerph-20-03474-t001:** Sociodemographic characteristics of the study participants (*n* = 356).

Characteristics	*n*	%
Age (mean, SD)	59.3	16.0
Gender (female)	243	68.3
Years living in Australia (mean, SD)	12.8	9.2
Education		
Primary school	29	8.2
High school/TAFE	136	38.4
University	189	53.1
English proficiency		
Excellent	5	1.4
Very good	33	9.1
Good	52	14.4
Fair	95	26.2
Poor	177	48.9
Employment status		
Full/part time	121	34.0
Retired or disability	233	66.1
Living with family	282	87.6
Health literacy		
Adequate	184	51.7
Inadequate/marginal	172	48.3
eHealth literacy (range 8–40)	24.8	8.0
Number of devices in use (mean SD)	2.3	1.3
Number of chronic conditions (mean SD)	1.0	1.2
Chronic conditions	*n*	**%**
Arthritis	106	29.9
Back pain	64	18.0
Cardiac conditions	57	16.1
Diabetes	44	12.4
Hypertension	30	8.5
Cancer	14	3.9
Depression and anxiety	14	3.9
Asthma	13	3.7
Stroke	6	1.7
Health status (self-reported)		
Excellent	2	0.6
Very good	44	12.2
Good	108	30.0
Fair	169	46.9
Poor	37	10.3

SD: standard deviation.

**Table 2 ijerph-20-03474-t002:** Mean scores for survey items in eHEALS * (*n* = 356).

Survey Items	Mean (SD)
I know what health resources are available on the Internet	3.21 (1.06)
I know how to use the health information I find on the Internet to help me	3.21 (1.10)
I know how to find helpful health resources on the Internet	3.17 (1.11)
I know where to find helpful health resources on the Internet	3.14 (1.06)
I know how to use Internet to answer my questions about health	3.08 (1.08)
I can tell high quality health resources from low quality health resources on the Internet	3.03 (1.11)
I feel confident in using information from the Internet to make health decisions	3.00 (1.10)
I have the skills I need to evaluate the health resources I find on the Internet	2.99 (1.10)

* Each of the 8 items were rated on a 5-point Likert scale, with 1 = strongly disagree and 5 = strongly agree. Overall eHealth Literacy Scale score ranges from 8 to 40.

**Table 3 ijerph-20-03474-t003:** Predictors of eHealth literacy of Chinese immigrants (adjusted) *.

Variable	Unstandardized Coefficients B	95% CI	Standardized Coefficients Beta	*p* Value
Age	−0.093	−0.154, −0.031	−0.175	0.003
Health status (5-point Likert scale from excellent to poor)	−1.544	−2.592, −0.496	−0.157	0.004
Number of technological devices used	1.103	0.404, 1.801	−0.173	0.002
Education level (primary/secondary/university)	3.088	1.753, 4.423	0.235	<0.001

* Linear regression model adjust for gender, English proficiency, and health literacy. CI: confidence interval (R = 0.578, R^2^ = 0.334, adjusted R^2^ = 0.327, *p* < 0.001).

## Data Availability

The data presented in this study are available on request from the corresponding author.

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
