# Peer review of "Online Health Information-Seeking Behaviours and eHealth Literacy among First-Generation Chinese Immigrants"

_ijerph, 2023, doi:10.3390/ijerph20043474_

Round 1

Reviewer 1 Report

The authors present an interesting study on eHealth and its correlates in a Chinese immigrant sample in Australia. The topic is of definitely of relevance for society and readers of IJERPH would likely be interested to learn more about these findings. The statistics appear plausible, and the manuscript is generally concise and well-written. However, I think some more efforts are required to make this a strong contribution. I will present some suggestions below.

I am concerned that continuous scales have been (arbitrarily) split into discrete categories which usually implies a loss of statistical power. Further, the concept of literacy is dimensional, and people should not be classified as good vs. poor unless the criterion has been validated (e.g., using ROC analyses using meaningful discrete groups). The same applies for the other scales (e.g., language proficiency, health literacy, etc.). The reporting of proportions is difficult to understand unless the categories have been shown to be meaningful.   

Generally, I believe it is suboptimal to use self-reports of proficiency rather that objective performance measures. It can be assumed that all proficiency scales are related: First, abilities usually show communality. Second, there may be a common reporting bias. In any case, I would prefer to see a correlation table for the dimensional scales. Next, a multiple regression analysis could follow (like the one given in the paper) to determine unique relationships. As the dependent variable, (continuous) eHealth could be predicted by the relevant other variables. Maybe a stepwise procedure could be used to identify the strongest predictors in a parsimonious model.

There appear to be particularities in the current sample that should be discussed: Generally, the sample is highly educated (53% completed university) and they have been living in Australia for a long time (12.8 years). In spite of this, a large proportion reports to speak poor English (75%) and they revealed poor health literacy (48%). On the one hand, it could be argued that Chinese people possibly segregated which could contribute to poor English proficiency. However, ca. 13 years of residency are a long time. And migrants who hold a university degree usually have proper health literacy. Further, they could also use Chinese websites to retrieve additional information. Maybe there was a reporting bias as well towards more modest responses?  

It could be discussed that the validity of eHealth may be investigated in future studies as a predictor of objectively assessed health. And health behavior could be tested as a mediator of this relationship.

Minor Issues

- Tables 2 and 3: The number of participants and the variance accounted for should be given in a note  below the table rather than in the title.

- Figure 2: I am wondering if the order of the categories corresponds with the order of the bars. The condition corresponding with poorer eHealth literacy is given first, but the first bar denoting the eHealth score is longer in most cases.

- Table 3: the 95% CI corresponds with the unstandardized B coefficient. Hence, it makes sense to switch its column with that of the standardized beta.

Reviewer 2 Report

The abstract is very simple and does not present information that generates interest in the reader. The abstract presents information about the cultural barriers, immigrants often have limited access to health information. Also, it includes percentages obtained with Lineal Regression models. However, it does not include a description of how to address the problem of online health information-seeking behaviors and eHealth literacy among first-generation Chinese immigrants and a brief description of the obtained results. This is very important in order to do more attractive the paper and facilitate the reader to read the rest of the manuscript.

 The introduction section does not describe the main contribution of the research presented. It is also necessary to include information on what the subsequent sections/content are about.

 Section 2. Materials and Methods lacks an introductory paragraph.  I suggest including a diagram or schema that depicts the steps or phases carried out to get the results. Also, I suggest consider to add a section or subsection of experiment design as part of Section 2.

On what basis did you consider the characteristics of the participants in the Results Section? It is necessary a justification

 In the Results, the authors mentioned the following: "In total, 415 surveys were received, of which 59 were incomplete and could not be 136 used". How long did it take to obtain the results? Do you consider that this sample or number of participants is sufficient to generate conclusive results? If yes, why?

The conclusions are very short and do not represent a relation with the obtained results. It is necessary to describe in more detail the conclusions. Additionally, future work is missed in the paper.

 The author Ling Zhang (Zhang, L.), self-cites 4 times in the paper. This is not considered ethical unless imperatively necessary and justified. Perhaps 2 self-citations would be considered appropriate. Please justify the reason and reduce the number of self-citations included in the paper.

Round 2

Reviewer 2 Report

The authors have improved the content of this paper and they have addressed the majority of the comments in the first review, but there are some points that  were not addressed properly,, for example:

1)      The introduction section does not describe the main contribution of the research presented. It is also necessary to include information on what the subsequent sections/content are about. The authors have introduced one paragraph describing in a brief way the main contribution.

2)      Section 2. Materials and Methods lacks an introductory paragraph.  I suggest including a diagram or schema that depicts the steps or phases carried out to get the results. Also, I suggest consider to add a section or subsection of experiment design as part of Section 2. I suggest presenting a PRISMA DiagraM to depict the phases of the methodological process,

3)      The conclusions are very short and do not represent a relation with the obtained results. It is necessary to describe in more detail the conclusions. Additionally, future work is missed in the paper. The authors have rewritten this section with the same idea. 
